# Depressive Symptoms among Individuals Hospitalized with COVID-19: Three-Month Follow-Up

**DOI:** 10.3390/brainsci11091175

**Published:** 2021-09-05

**Authors:** Paolo Vassalini, Riccardo Serra, Lorenzo Tarsitani, Alexia E. Koukopoulos, Cristian Borrazzo, Federica Alessi, Chiara Di Nicolantonio, Cecilia Tosato, Francesco Alessandri, Giancarlo Ceccarelli, Claudio Maria Mastroianni, Gabriella d’Ettorre

**Affiliations:** 1Department of Public Health and Infectious Diseases, Policlinico Umberto I, Sapienza University of Rome, Piazzale Aldo Moro 5, 00185 Rome, Italy; cristian.borrazzo@uniroma1.it (C.B.); federica.alessi@uniroma1.it (F.A.); ceci.tosato@gmail.com (C.T.); giancarlo.ceccarelli@uniroma1.it (G.C.); claudio.mastroianni@uniroma1.it (C.M.M.); gabriella.dettorre@uniroma1.it (G.d.); 2Department of Human Neurosciences, Policlinico Umberto I, Sapienza University of Rome, Viale dell’Università 30, 00185 Rome, Italy; riccardo.serra91@gmail.com (R.S.); lorenzo.tarsitani@uniroma1.it (L.T.); alexiakoukopoulos@gmail.com (A.E.K.); chiadin@gmail.com (C.D.N.); 3Intensive Care Unit, Department of General and Specialist Surgery “Paride Stefanini”, Policlinico Umberto I, Sapienza University of Rome, Piazzale Aldo Moro 5, 00185 Rome, Italy; francescoalessandri1@yahoo.it

**Keywords:** COVID-19, SARS-CoV-2, depression, mental health, hospitalization

## Abstract

Individuals affected by Coronavirus Disease 2019 (COVID-19) may experience psychiatric symptoms, including depression and suicidal ideation, that could lead to chronic impairment and a reduction in quality of life. Specifically, depressive disorder shows high incidence and may lead to chronic impairment and a reduction in the quality of life. To date, no studies on the presence of suicidality and quantitative analysis of depressive symptoms and their risk factors have yet been published. In this study, we aim to assess the prevalence of depressive symptoms and related risk factors at 3 months after discharge to home care following hospitalization for COVID-19 infection. Methods: Participants were contacted three months after hospital discharge from one of the five COVID-19 hospitals in Rome, as part of a larger project on health outcomes in COVID-19 inpatients (Long Term Neuropsychiatric Disorder in COVID-19 Project), and the Patient Health Questionnaire-9 (PHQ-9) was administered by telephone interview. Results: Of 115 participants, 14.8% (N = 17) received a PHQ-9-based diagnosis of depression, and *n* = 7 of them scored 1 or more on the item on suicidality. A linear regression model showed the predictive role of female sex, pulmonary chronic condition and previous mental disorder in the development of depressive disorder; the latter was confirmed also by binary logistic regression. Severity indexes of disease (length of hospitalization and intensive care treatment) were found not to be associated with the development of depressive symptoms. Conclusions: A small but clinically meaningful number of participants in the current study reported that they experienced symptoms of depression and suicidal ideation 3 months post-discharge from their COVID-19 hospitalization. In particular, given the findings that a history of prior psychiatric disorders was predictive of the development of depression symptoms, clinicians should carefully monitor for the presence of all psychiatric symptoms at follow-up visits.

## 1. Introduction

Since December 2019, the novel Severe Acute Respiratory Syndrome Coronavirus 2 (SARS-CoV-2), the cause of the Coronavirus Disease 19 (COVID-19), has rapidly spread around the world [1]. In March 2020, Italy was severely affected by the epidemic and has the second-largest number of confirmed COVID-19 cases, after China [2].

Clinical manifestations of COVID-19 range from an asymptomatic disease to acute respiratory distress syndrome (ARDS), multiorgan failure and shock and can lead to death in up to 25–62% of critically ill patients [3,4,5,6]. Evidence has shown that it is crucial to arrange follow-up visits for discharged individuals in order to quickly identify possible organ damages and long-term sequelae [7].

Previous data on severe acute respiratory syndrome (SARS) infection showed that coronavirus is associated with sustained mental disorders and long-lasting neuropsychiatric consequences [8]. Specifically, COVID-19 individuals have shown symptoms of psychological distress that may lead to chronic impairment and a reduction in their quality of life; emerging evidence has also shown that, following hospitalization, these individuals are at high risk of developing Post-Traumatic Stress Disorder [9,10].

A few studies have focused on the relation between COVID-19 and depression during and after hospitalization [11,12,13]. These studies showed that symptoms of depression are highly prevalent during the hospital stay and tend to decrease after discharge. One study has reported data on the long-term prevalence of the subjective presence of depressive symptoms [14]. Although powerful in consideration of the number of participants involved, this study relies on one item, a 0–4 scale indicating the self-reported level of anxiety or depression. To our knowledge, no studies on the presence of suicidality and quantitative analysis of depressive symptoms and their risk factors have yet been published. The aim of this study was to assess the prevalence of depressive symptoms and related risk factors at 3 months after discharge to home care following hospitalization for COVID-19 infection. The study was carried out in a large COVID-19 University Hospital in Rome.

## 2. Materials and Methods

### 2.1. Participants

Participants were recruited among those admitted to the Division of Infectious Diseases, at the Department of Public Health and Infectious Diseases of Umberto I “Sapienza” University Hospital of Rome, during the height of the pandemic in Rome. As one of the 5 COVID-19 hospitals of Rome, “Umberto I” is a 1200-bed teaching hospital, with a catchment area of 600,000–1,200,000 people [15]. This study is part of a larger project on health outcomes in COVID-19 inpatients, named “Long Term Neuropsychiatric Disorder in COVID-19 Project”. Inclusion criteria included: age > 18 years, hospitalization for a confirmed case of COVID-19 and discharge to home care. Recruitment was carried out over two months. Individuals with a clinically evident cognitive impairment, active mental disorders or inadequate knowledge of the Italian language were excluded. Participants with non-remitting COVID-19, or unrelated clinical conditions, involving being transferred to other hospital wards or medical facilities, were also excluded.

Between one and two days before discharge, participants were approached by the unit doctors and received a complete explanation of the purpose and procedures of the study and gave written informed consent to participate. Clinical and demographic variables were retrieved from clinical records. Sociodemographic variables, comorbidities, previous mental health conditions, treatment undergone during hospitalization and length of hospital stay were included in the dataset. Three months after discharge, trained clinical raters contacted the participants by telephone. The telephone call included a renewal of the informed consent and an interview including specific tools. Although a face-to-face assessment with a structured psychiatric interview would have been preferable, this methodology was the only one available during the height of the pandemic. Information about the study and its aims were provided again, and verbal consensus was requested to confirm the informed consent provided during hospitalization. If they consented, information about current clinical state and new medical treatments was retrieved. In order to gather specific data on depressive symptomatology, raters administered the Patient Health Questionnaire-9 (PHQ-9). Individuals with signs of clinically significant mental distress during assessment were offered to be referred to the Psychiatry Outpatient Service of our hospital or other second-level mental health services, as appropriate. The study received the approval of the Sapienza University of Rome Ethical Committee (Ref. 109/2020).

### 2.2. Instruments

PHQ-9 is a validated screening tool for assessing the presence and severity of depression in the clinical setting [16,17]. Participants are asked to rate, between 0 (never) and 3 (almost every day), how frequently they experience a variety of depressive symptoms as reported by the Diagnostic and Statistical Manual of Mental Disorders fifth edition (DSM-5) (Figure 1) [18]. If any symptom is reported, participants are asked to rate from 0 to 3 how much impact such symptoms have on their daily functioning (i.e., working, taking care of home and getting along with others). For the purpose of this study, we used the continuous variable “total score” and a 10-point cut-off indicating the presence of clinically relevant depressive symptomatology [19].

### 2.3. Statistical Analysis

Analysis was performed using Statistical Package for Social Science (SPSS) version 25 (IBM Corp. Released 2017. IBM SPSS Statistics for Windows, Version 25.0. Armonk, NY, USA: IBM Corp).

Continuous variables were summarized into either medians with interquartile range (IQR, 25th and 75th percentile) or into means with standard deviation (±SD), as appropriate, along with the related ranges (minimum–maximum). Categorical variables were summarized as frequencies (*n*), proportions or percentages and rates, as appropriate.

Univariate analyses were conducted for all available sociodemographic and clinical variables. In order to select potentially meaningful organic conditions, we built a stepwise regression model predicting PHQ-9 total score, including all the pre-existing chronic medical conditions as possible predictors of depression (obesity, immunodepression, hypertension, dysthyroidism, cardiopathy, diabetes, cancer, inflammatory bowel diseases and vascular pathologies) [20,21,22,23,24,25,26,27]. Obesity was considered as body mass index (BMI) > 30 (<18.5 underweight range, 18.5–24.9 healthy weight range, 25–29.9 overweight range, >30 obese range) [28]. A multivariate linear regression model was built to analyze possible predictors of the PHQ-9 total score and a binary logistic regression model was built to test for predictors of clinically relevant depressive symptomatology. Odds ratios (ORs) and 95% confidence intervals (95% CIs) were calculated for all associations. *p*-value < 0.05 was considered statistically significant.

## 3. Results

A total of 183 consecutive participants discharged to home care during the study time span were assessed for eligibility. Of these, 68 were excluded (*n* = 25 did not fulfil inclusion criteria, *n* = 46 were missed, *n* = 2 declined to participate). A total of *n* = 115 individuals were recruited.

No significant differences in sociodemographic variables were found between participants who were recruited and those lost during recruitment in age (median 57 (IQR 25–75%) 48–66 vs. 60.5 (IQR 25–75%) 48–72, respectively *p* = 0.200) or sex (male 54% vs. 42%, respectively *p* = 0.300).

At 3-month follow up, 14.8% of the sample (N = 17) received a PHQ-9-based diagnosis of depression (total score > 9). Mean PHQ-9 total score was 4.82 ± 0.79 (range= 0–25). *n* = 41 (35.65%) participants scored at least 1 on the item on subjective impairment, with *n* = 7 scoring 2 or more. A total of *n* = 7 participants scored 1 or more on the item on suicidality (these individuals were referred to secondary mental health services).

Demographic and clinical characteristics of the sample are presented in Table 1.

The stepwise regression model highlighted a potential role of chronic pulmonary conditions, over and above other chronic conditions (B = 4.50; SE = 1.26; *p* < 0.001). We therefore included anamnesis of chronic pulmonary conditions in the models. The linear regression model predicting PHQ-9 total score (Table 2) highlighted the predictive role of chronic pulmonary conditions, previous mental disorders and sex. On the other hand, the binary logistic regression model predicting a PHQ-9 diagnosis of depression (Table 3) confirmed the predictive role of previous psychiatric diagnosis only.

## 4. Discussion

In this observational study on COVID-19 survivors discharged after hospitalization for COVID-19 during the height of the pandemic in Rome, we analyzed quantitative aspects of depression and the presence of suicidality at a 3-month FU. More than one out of ten participants received a provisional diagnosis of depressive disorder and the average and more than one of three reported some degree of impairment caused by depressive symptoms. Concerningly, 6% of the individuals reported the onset of suicidality. Findings from this study confirm and deepen the emerging evidence from other studies on the topic and are in line with similar reports from past epidemics (e.g., the SARS epidemic in 2002–2004).

The etiology of depression following hospitalization due to infection with coronavirus is likely to be multifactorial, involving organic, psychological and social factors. On an organic level, causes might include the direct effects of the viral infection, cerebrovascular consequences, the immunological response and medical interventions. On a psychological level, we believe that the fear of a novel severe and potentially fatal illness, the following stigma and the isolation have played a role [29,30,31]. This is especially true in consideration of the domino effect that COVID-19 had, spreading rapidly within households and family members [32]. Our data highlight the role of previous psychiatric conditions and female sex, which are commonly known factors in the development of mental disorders in response to a variety of stressors [33]. Although further research is needed to understand the role of chronic pulmonary diseases, considering that chronic pulmonary disease was a widely known risk factor for COVID-19 severity, we hypothesize that it might have worked as a booster of fear (before, during and after hospitalization) and of prophylactic social isolation. This hypothesis would be in line with our data showing how, in our cohort, the presence of depressive symptomatology is not associated with variables indicating a more severe disease (i.e., duration of hospitalization, ICU treatment).

We hypothesize that social factors played a role in the development of depressive symptoms: grief, fears of loss and stigma, as well as the repeated exposure to stressors, such as the restrictive measures implemented nationwide (e.g., full lockdown) and the daily media reports on local and global consequences of COVID-19. Indeed, evidence from different conditions has shown that these factors in their complexity might increase vulnerability to depressive disorder [34]. Moreover, the economical and labor crisis should also be taken into account in consideration of the literature supporting their relation with depression and anxiety [35]. Regarding this hypothesis, it is interesting to note how social support, which, in many cases, was drastically reduced due to restrictions, seems to play a moderating role in the relation between economic stressors and depression [36]. Further research is needed to test these hypotheses and attempt to uncover the complex pathogenesis of depression following severe COVID-19 infection.

### Limitations

The telephone interview methodology is the major limitation of our study. Although this method has been used successfully in previous research during the current pandemic and raters were trained in interview skills before this study, a face-to-face assessment with a structured psychiatric interview would have been preferable [37,38]. A face-to-face interview, indeed, would offer the chance to perform cross-checking with visible symptomatology and investigate possible impairments in daily functions. Despite this, the methodology that we used was the only one available during the height of the pandemic and in a setting where part of the population was not able to use a virtual platform.

## 5. Conclusions

A small but meaningful number of COVID-19 individuals experience some discomfort due to depressive symptoms and clinically relevant depressive symptomatology after hospitalization, developing, in some cases, suicidal ideation. Further research, including more complete assessment and face-to-face psychiatric interviews, is needed to understand the complex etiology of this long-term consequence. Clinicians should consider evaluating the presence of depression and of suicidal ideation in follow-up control visits for COVID-19 individuals.

## Figures and Tables

**Figure 1 brainsci-11-01175-f001:**
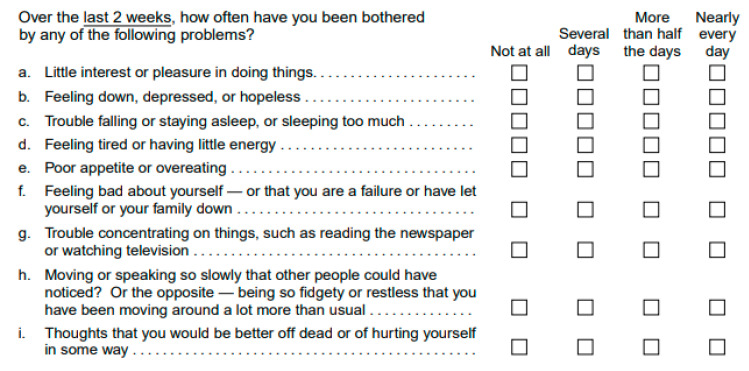
Patient Health Questionnaire-9 (PHQ-9).

**Table 1 brainsci-11-01175-t001:** Demographic and clinical characteristics of the population.

	All Sample (*n* = 115)
Sex Male, *n* (%) Female, *n* (%)	62 (54) 53 (46)
Age, median (IQR), years ≥65 years, *n* (%)	57 (48–66) 29 (25)
Lenght of hospital stay, median (IQR), days	15 (10–23)
Intensive care treatment, *n* (%)	26 (23)
Previous psychiatric diagnosis, *n* (%)	9 (8)
Obesity (BMI > 30), *n* (%)	5 (4)
Chronic pulmonary diseases ^a^, *n* (%)	15 (13)

^a^ Including chronic obstructive pulmonary disease and asthma; IQR: interquartile range. BMI: body mass index.

**Table 2 brainsci-11-01175-t002:** Linear regression model predicting PHQ-9 total score.

	B	SE	*p*-Value	95% CI
Chronic pulmunary diseases ^a^	4.733	1.082	<0.001	2.589–6.877
Previous psychiatric diagnosis	6.354	1.395	<0.001	3.589–9.118
Sex (male)	−3.098	0.769	<0.001	−4.622–−1.574
Age	−0.05	0.029	0.092	−0.108–0.008
Lenght of hospitalization (days)	0.025	0.043	0.557	−0.06–0.11
Non-intensive care	0.498	0.81	0.540	−1.107–2.103

^a^ Including chronic obstructive pulmonary disease and asthma.

**Table 3 brainsci-11-01175-t003:** Binary logistic regression model predicting PHQ-9 based diagnosis of depression.

	B	SE	*p*-Value	OR	95% CI
Chronic pulmunary diseases ^a^	1.320	0.727	0.069	3.745	0.900–15.582
Previous psychiatric diagnosis	2.674	0.827	0.001	14.502	2.869–73.305
Sex (male)	−1.117	0.632	0.077	0.327	0.095–1.129
Age	−0.014	0.024	0.566	0.986	0.940–1.034
Lenght of hospitalization (days)	−0.016	0.037	0.661	0.984	0.916–1.057
Non-intensive care	0.611	0.661	0.356	1.842	0.504–6.728

^a^ Including chronic obstructive pulmonary disease and asthma.

## Data Availability

The raw data and materials used for this study are available upon request.

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
