# Peer review of "Depressive Symptoms among Individuals Hospitalized with COVID-19: Three-Month Follow-Up"

_brainsci, 2021, doi:10.3390/brainsci11091175_

Round 1

Reviewer 1 Report

Brainsci-1358246 Reviewer Comments

Title: *Depressive Symptoms Among Covid-19 Survived Patients: A 3 Month Follow-Up

Reviewer Comments:

Summary of Manuscript:

This study aimed to assess the prevalence of depressive symptoms and related risk factors of 115 participants at a 3 month follow-up after their hospital discharge to home care from COVID-19. Participants were assessed via phone interviews utilizing the Patient Health Questionnaire-9 (PHQ-9), with 14.8% (n = 17) receiving a related diagnosis of depression, and n = 7 participants scoring 1 or more on the suicidality item. Linear regression showed that female sex, chronic pulmonary conditions, and previous psychiatric disorders, were predictive of depression; binary logistic regression indicated, however, that only previous psychiatric disorders were predictive.

Severity indices of COVID-19 (length of hospitalization, and intensive care treatment) were found not to be associated with the development of depressive symptoms. The authors conclude that a large proportion of individuals with COVID-19 experience some depressive symptoms after hospitalization, a considerable number develop major depression, and in some cases, suicidal ideation. They suggest that clinicians should assess for the presence of depression and suicidal ideation in follow-up visits for individuals that were hospitalized for COVID-19.

*I acknowledge and appreciate the fact that the authors’ native language is Italian, and overall the manuscript reads very well. There are a few areas where the phrasing is a bit awkward, though, and I have made some note/suggestions for rephrasing in the sections below.

*The phrase “survived patients” is a bit awkward. I suggest rephrasing the title to:

“Depressive Symptoms Among Individuals Hospitalized with COVID-19: 3 Month Follow-Up”

Overall Impression:

This manuscript adds to the literature by providing COVID-19 post-hospitalization 3-month follow-up assessments of depressive symptoms and suicidality. Interestingly, severity indices of COVID-19 (length of hospitalization, and intensive care treatment) were not associated with the development of depressive symptoms, which also contributes to the study findings. Overall, the manuscript is worthy of publication, however, it requires minor revisions before it is accepted. Please see below for specific comments in each area.

*Title:

*See my note above re: suggested rephrasing of the title.

Abstract:

Overall, the abstract provides the rationale for the study. However, there are a few points which should be revised. Please consistently use the term “participants” or “individuals,” vs “patients” or “subjects,” throughout the abstract and manuscript. Please also consistently capitalize the first letter of the word beginning the sentence after the heading (e.g., “Abstract: patients…”), and have space between the headings (e.g., “infection.Methods: patients…”). Here and throughout the manuscript, please be clear about the distinctions between what you are measuring (e.g., “depressive symptoms,” vs “Major Depressive Disorder (MDD)”). I would also specify that the participants were interviewed via self-report over the phone after discharge from one of the 5 COVID-19 hospitals of Rome, as part of larger project on health outcomes in COVID-19 inpatients [name of the study]. It is a bit misleading to indicate that you are collecting “qualitative data” in the current study, as it is really self-report quantitative ratings of symptoms and degrees of impairment, vs. participants relating individual descriptions of their experiences, with the raters performing thematic analysis to report on qualitative summary descriptions. Also, I would temper the language that you use in the abstract aims and conclusions sections about all individuals with COVID-19 experiencing psychiatric symptoms:

Current Version:

“Abstract: patients affected by coronavirus disease 2019 (COVID-19) experience long-lasting sustained mental disorders and neuropsychiatric consequences. Specifically, depressive disorder shows high incidence and may lead to chronic impairment and reduction of the quality of life…” “Conclusion: A large proportion of COVID-19 patients experience some discomfort due to depressive symptoms after hospitalization, a considerable number develop major depression, and in some case suicidal ideation...”

Suggested Revision:

Abstract: Individuals affected by Coronavirus 2019 (COVID-19) may experience psychiatric symptoms, including depression and suicidal ideation, that could lead to chronic impairment, and reduction of quality of life. To date…”

Conclusion: A small but clinically meaningful number of participants in the current study reported that they experienced symptoms of depression and suicidal ideation 3 months post-discharge from their COVID-19 hospitalization. Particularly given the findings that a history of prior psychiatric disorders was predictive of the development of depression symptoms, clinicians should carefully monitor for the presence of all psychiatric symptoms at follow-up visits.”

Keywords:

The keywords are accurate for the topic of the manuscript. This is a minor point, but the authors may want to group the “depression,” “mental health” and “hospitalization” keywords together, and the related “Sars-COV-2” and “COVID-19” together in the list order for organization.

Introduction:

The authors provide a succinct overview of the literature with excellent recent references, leading up to their rationale for the current study. However, please see my notes above for the abstract that also apply to this section. Additionally, here and throughout the manuscript, please spell out acronyms at first use, and then consistently use them afterwards (e.g., SARS=CoV-2, COVID-19).

Materials and Methods:

Overall, the materials and methods section is outlined very thoughtfully. In addition to my notes above that also apply to this section, there are other points to note. Please add citations for SPSS version 25, and all references for the PHQ-9 (actual instrument, validation studies, clinical and research setting uses, rationale for 10 points clinical cut-off). Given that the PHQ-9 is the only (self-report) instrument that is used in this study (which I would add to the limitations), I suggest adding another table to specifically note all of the PHQ-9 question for the reader.

Awkward phrases that could be revised include “In order to avoid overestimation…were also excluded.” I would rephrase to “Participants with non-remitting COVID-19, or unrelated clinical conditions, involving being transferred to other hospital wards or medical facilities, were also excluded.” In the statistical analysis section, the run-on sentence “Continuous variables…” could be rephrased to “Continuous variables were summarized into either medians with interquartile ranges (IQR; 25th and 75th percentiles), or into means with standard deviations (±SD), as appropriate, along with the related ranges (minimum-maximum). Categorical variables were summarized as frequencies (n), proportions or percentages, and rates, as appropriate.” The phrase “meaningful organic conditions” is not well-recognized in the USA. I suggest rephrasing to “pre-existing medical conditions,” and then adding literature citations to provide the rationale for why you are selecting these conditions/variables as your predictors. I would also describe the related measurements and spell out any acronyms (e.g., for obesity, which organization/reference are you utilizing? Please spell out BMI at first use, and briefly give the ranges so that readers know that obesity is defined in this case as a BMI > 30). This is also the place to describe in more detail the procedure for conducting the phone interviews; it is not made clear until the discussion currently that this was the method (vs. in person), which can be very misleading to the readers.

Results:

The presentation of the results is succinct. In addition to my notes above that also apply to this section, there are a few other points to note. For ease of readability, please bold font the references to the tables in the text, as well as in the table titles, so that they stand out. Please ensure that the findings are presented in complete sentences (e.g., “Age…and sex…”), and that the spelling is correct throughout (e.g., “Lenght* of hospitalization”). Please include footnotes for all tables to include related references, and spell out acronyms, as noted above. Each table should be able to stand on its own for readers without reference to the other tables or the text.

Discussion:

The authors give a nice summary of the study findings and tie it in to the existing literature. Please see all of my notes above that also apply to this section. As I mentioned before, there are several points that are currently made in the discussion section for the first time that should be made in prior relevant sections (see my notes above). The fact that the study was conducted during the “hot phase” of the pandemic in Rome should be mentioned previously. The authors’ hypothesis that “potential griefs, fears of loss and stigma might also play a role in the *aetiology of depression in these patients” is interesting, along with factors related to full lockdown, media, economic and labor crisis, and lack of social support. There is a lot of information here, and it may be helpful for the reader to expand on these points to further tie them all together.

*”Aetiology” is spelled as “etiology” in the USA.

Limitations:

The authors correctly point out that a main limitation of the study was the use of telephone vs in person assessments. While it may seem obvious, it is worth reaffirming to the reader that this method was the only available one during the height of the pandemic (assuming that Zoom or other virtual platforms were not an option; if so, I would specifically state that). Another major limitation of the study, though, as I had pointed out, is the reliance on one self-report measure. This means that the raters are wholly dependent on what the participants are reporting to them, with no cross-checks of visible symptomatology, or impairments in daily functioning. I would add this to the limitations, and suggest building this into future studies when the severity of the pandemic eases, and allows for further in-person follow ups with cross-checks of the data.

Conclusions:

As I noted above, the conclusions section should be focused on the specific results of this study, vs an over-generalizing of findings of all patients with COVID-19. The authors can add to the points about future research with some of the suggestions that I have made above.

Tables:

*As noted above, please bold font the mentions of all tables in the text so that the reader can more clearly see and refer to them. Please also consider adding at table outlining the PHQ-9.

Back Matter: These sections of the manuscript seem to be provided per Brain Sciences:

Author contributions

Funding

Institutional review board statement

Informed consent statement

Conflicts of interest

References

Back Matter: These sections are not provided in the version of the manuscript that I have:

Data Availability Statement

Acknowledgments

Reviewer 2 Report

The authors of the manuscript ‘Depressive Symptoms Among Covid-19 Survived Patients: A 3 Month Follow-Up’  assessed depressive symptoms in Covid-19 survived patients over 3 months of follow-up.

This was a phone screening-based and questionnaire-based cross-sectional clinical study where the authors contacted Covid-19 survivors after 3 months from hospital discharge. The study included both males and females, and gender analysis is included. The authors further used chronic pulmonary condition, psychiatric condition, length of hospital stay as factors to assess the trend in the occurrence of depression and severity of depression using the PHQ-9 scale.  The data were analyzed using a linear regression model and binary logistic regression model to understand the interaction and prediction of depression in the presence of a chronic pulmonary condition or psychiatric conditions. 

This is a very well-designed, conducted, and implemented the clinical study. Outcome measures and analysis support the hypothesis. The data is presented clearly in the tables. Considering the concise format and small number of subjects it can be presented as a short report.
